# Potential Antioxidant and Neuroprotective Effect of Polysaccharide Isolated from Digüeñe *Cyttaria espinosae*

**DOI:** 10.3390/jof11090637

**Published:** 2025-08-29

**Authors:** Claudia Pérez, Fabián A. Figueroa, Ignacio Tello, Roberto T. Abdala-Díaz, Manuel Marí-Beffa, Viviana Salazar-Vidal, José Becerra, Javiera Gavilán, Jorge Fuentealba

**Affiliations:** 1Laboratorio de Química de Productos Naturales, Departamento de Botánica, Facultad de Ciencias Naturales y Oceanográficas, Universidad de Concepción, Concepción 4030000, Chile; fabian.figueroa@ucsc.cl (F.A.F.); itello2016@udec.cl (I.T.); vivianasalazar@udec.cl (V.S.-V.); jbecerra@udec.cl (J.B.); 2Facultad de Ciencias, Universidad Católica de la Santísima Concepción, Concepción 4090541, Chile; 3Departamento de Ecología y Geología, Facultad de Ciencias, Universidad de Málaga, E-29071 Málaga, Spain; abdala@uma.es; 4Instituto Andaluz de Biotecnología y Desarrollo Azul (IBYDA), Centro Experimental Grice Hutchinson, Universidad de Málaga. Lomas de San Julián, 2, 29004 Málaga, Spain; beffa@uma.es; 5Departamento de Biología Celular, Genética y Fisiología, Universidad de Málaga, E-29071 Málaga, Spain; javieragavilan@udec.cl; 6Laboratory of Screening of Neuroactive Compounds, Department of Physiology, Faculty of Biological Sciences, University of Concepción, Concepción 4030000, Chile

**Keywords:** β-amyloid peptide, neuroprotection, *Cyttaria espinosae*, fungal polysaccharides

## Abstract

Alzheimer’s disease (AD) is a significant global health challenge, further exacerbated by the anticipated increase in prevalence in the coming years. The accumulation of β-amyloid peptide plays a critical role in the onset of AD; however, emerging evidence suggests that soluble oligomers of β-amyloid may primarily drive the neuronal impairments associated with this condition. Additionally, neurodegenerative diseases like AD are linked to oxidative stress and reduced antioxidant capacity in the brain. Natural products, particularly polysaccharides extracted from mushrooms, have garnered interest due to their neuroprotective properties and the potential to enhance the value of natural sources in addressing human diseases. This study examines the antioxidant and neuroprotective properties of polysaccharides derived from *Cyttaria espinosae* Lloyd (CePs), a relatively underexplored fungus native to Chile. Using Fourier Transform Infrared Spectroscopy (FT-IR) and Gas Chromatography-Mass Spectrometry (GC-MS), we characterized CePs. We assessed their antioxidant capacity using DPPH and ABTS assays, yielding maximum inhibition rates of 32.14% and 19.10%, respectively, at a concentration of 10 mg mL^−1^. CePs showed no toxicity in zebrafish embryos and maintained high cell viability in PC-12 cells exposed to amyloid β peptide (Aβ). Our findings suggest that CePs exhibit significant antioxidant and neuroprotective properties against Aβ peptide toxicity while remaining non-toxic to zebrafish embryos. This underscores the potential of the polysaccharides from this mushroom to serve as functional foods that mitigate oxidative stress and warrant further investigation into their mechanisms in the context of the physiopathology of Alzheimer’s disease.

## 1. Introduction

Macrofungi, which include representatives of the Basidiomycota and Ascomycota phyla, are distinguished by their fruiting bodies, which can be observed without a microscope [1,2]. Approximately 14,000 macrofungi species have been identified globally, with 2000 being edible and medicinal [1]. At least 270 species have been investigated for their secondary metabolites and potential medicinal applications [2]. The chemical composition of these mushrooms is primarily water, with carbohydrates representing most of the dry matter, followed by proteins and a low percentage of lipids [3]. Furthermore, these organisms produce a variety of compounds with pharmacological effects, including polysaccharides, unsaturated fatty acids, biologically active proteins, phenolic compounds, vitamins, terpenoids, ergosterols, and volatile organic compounds [4].

The pharmacological potential of mushroom compounds has led some scientists to consider them potential treatments for several diseases, including cancer, immune disorders, and neurodegenerative diseases [5]. The medicinal potential of these organisms in addressing neurodegenerative diseases is well-documented in the literature [6,7,8,9]. *Cordyceps cicadae* is a medicinal mushroom that has been used since the 5th century BC [10]. It has been demonstrated to have potent neuroprotective activity against glutamate-induced damage in PC-12 cells, with a 35.1% increase in cell viability, a 36.6% inhibition of lactate dehydrogenase enzyme activity, and a 15.8% reduction in cell apoptosis at a concentration of 200 µg mL^−1^ [11]. Furthermore, *Cordyceps militaris*, another fungus belonging to the same genus, contains a chemical compound designated as cordycepin, which has been demonstrated to enhance locomotor abilities in mice afflicted with Parkinson’s disease and to protect dopaminergic neurons from inflammatory damage induced by lipopolysaccharides (LPS) [12]. The mycelium of *Hericium erinaceus*, a mushroom with culinary and traditional Chinese medicinal uses, exhibits strong antioxidant properties, achieving up to 90% inhibition in the DPPH assay at 1000 µg mL^−1^, like ascorbic acid’s 100% inhibition at the same concentration. This research shows that mycelium protects PC-12 cells from *N*-Methyl-4-phenylpyridinium (MPP^+^), a neurotoxin, increasing cell survival by 40%. In contrast, aqueous extracts of this fungus can reverse a 30% decline in cell viability, protect the mitochondrial membrane, and alleviate Ca^2+^ overload, with ROS levels induced by glutamate decreasing by 90% in PC-12 cells [13].

The genus *Cyttaria* Berk. is classified within the division Ascomycota, class Leotiomycetes, order Cyttariales, and family Cyttariaceae. A total of 11 species belonging to this genus have been described worldwide, with seven occurring in South America and four in Oceania [14]. In Chile, members of this genus are commonly known as digüeñes and are distributed from the central-southern area to Patagonia. They exclusively parasitize the *Nothofagus* genus. Most research on the genus *Cyttaria* has focused on its taxonomy, evolution, and ecology. Nevertheless, only a limited number of studies have been conducted on the chemical characteristics of the *Cyttaria espinosae* (*C. espinosae*), with a particular focus on its nutritional parameters, cytotoxic activity, and antioxidant capacity [15]. Regrettably, these studies have not explored the potential applications of the species in this genus for treating neurodegenerative diseases.

This study aims to assess the antioxidant and neuroprotective effects of CePs on amyloid beta (Aβ)-induced neurotoxicity in neuronal-like cells while also evaluating the in vivo toxicity of this polysaccharide in zebrafish embryos. The findings suggest that this mushroom has, in addition to nutritional benefits, radical scavenging and neuroprotective properties against the Aβ peptide toxicity, which is, in parallel, non-toxic to zebrafish embryos. Thus, the polysaccharides of this genus can be utilized as functional foods to mitigate oxidative stress and explore new chemical spaces for studying strategies against Aβ toxicity.

## 2. Materials and Methods

### 2.1. Materials and Chemicals

Standard monosaccharides, including L-arabinose (Ara), erythrose (Ery), L-fucose, D-galactose (Gal), D-glucose (Glc), D-mannose (Man), L-rhamnose (Rha), D-ribose (Rib), and D-xylose (Xyl), were obtained from Sigma Chemical Co. (St. Louis, MO, USA). All monosaccharides were of analytical grade with a purity ≥99.5% (GC), as specified by the supplier. Additionally, chemical reagents were obtained from Sigma–Aldrich (St. Louis, MO, USA). Standards of gallic acid (≥98% purity), Trolox (≥98% purity, titration), 1, 2, 2-diphenyl-1-picrylhydrazyl (DPPH^•+^, ≥95% purity), and ascorbic acid (≥99% purity) were of analytical grade and used without further purification. Ethanol (≥99.8%, analytical grade), methanol (≥99.9%, analytical grade), and the Folin–Ciocalteu reagent were supplied by Merck (Darmstadt, Germany).

### 2.2. Collection and Identification of C. espinosae

#### 2.2.1. Collection

The fruiting bodies of *C. espinosae* were collected during the 2017–2018 spring-summer season in the Nahuelbuta area, specifically in the Palos Quemados sector of the Biobío Region, Chile. The *C. espinosae* specimens were fully mature, exhibiting an orange-ochre color and measuring 5.2 ± 0.3 cm in size. Morphologically, they had a fleshy, globose shape, formed by spherical swellings on *Nothofagus* species. After collection, the stromata were transferred to the Natural Products Chemistry Laboratory at the University of Concepción. For reference, voucher specimens were deposited in the Fungario de ONG Micófilos (MICOCL).

#### 2.2.2. Identification

In the identification of *C. espinosae* samples, we carefully documented the macroscopic characteristics of the stromata in situ, utilizing photographic documentation via a Canon T6 reflex camera for both field and laboratory observations. This step is vital for establishing a reliable morphological baseline for identification purposes. Taxonomic determination was then achieved through an examination of microscopic features. Stromata morphology was examined using a Motic BA210S Trinocular compound microscope (Barcelona, Spain) with a Moticam S6 camera attached. Microscopic structures were studied by mounting tissue sections of dried stromata in demineralized water and lactophenol cotton blue according to established methods [16].

After identification, *C. espinosae* stromata were carefully labeled and frozen at −20 °C. Then, 108 specimens were randomly divided into three groups of 36 each. Each group was independently freeze-dried for 48 h using a CHRIST Alpha 2–4 LO plus automatic freeze-dryer (Martin Christ, Osterode am Harz, Germany). After freeze-drying, the material from each group was pooled to create three replicates (*n* = 3) of freeze-dried material, which served as independent samples for further analysis. The freeze-dried samples were stored at −20 °C to ensure long-term preservation, prevent ice crystal formation that could damage the structure, and maintain their bioactive properties for future studies [17,18].

### 2.3. Isolation and Purification of Fruiting Body Polysaccharides

Freeze-dried samples of *C. espinosae* were ground using a Moulinex DP800 grinder (Moulinex, Bagnolet, France), and the initial mass was recorded. Polysaccharides were extracted using the method described in previous studies with minor modifications [19,20]. Before extraction, the biomass was suspended in 70% ethanol (EtOH) for 48 h with continuous agitation. The solid residue was dried at 25 °C and then resuspended in 500 mL of distilled water, which was heated at 100 °C for one hour. The solution was centrifuged at 4500 rpm for 5 min to recover the supernatant. This procedure was repeated twice for thorough extraction.

The collected supernatant was concentrated using a rotary evaporator (IKA HB10 digital, Staufen, Germany) at 90 rpm under reduced pressure and at 65 °C until its volume was reduced to 400 mL. To precipitate the polysaccharides, absolute EtOH was added in a 1:1 ratio to the supernatant and kept at 4 °C for 24 h. The sample was then centrifuged at 4500 rpm for 15 min to collect the CePs, which were subsequently lyophilized using an automatic freeze-dryer (model CHRIST Alpha 2–4 LO plus) and stored at −20 °C for future use.

For additional purification, crude polysaccharides were dissolved in a 4 M NaCl solution, after which 99% EtOH was added and stored at 4 °C for 24 h. The solution was centrifuged at 4500 rpm for 15 min, and then the resulting mixture was dialyzed under low osmotic pressure (1 M NaCl). Finally, the solution underwent lyophilization in the CHRIST Alpha 2–4 LO plus automatic lyophilizer, yielding the CePs.

### 2.4. Elemental Analysis of CePs

The total content of Carbon (C), Hydrogen (H), Nitrogen (N), and Sulfur (S) in CePs was analyzed using a LECO TruSpec Micro CHNSO elemental analyzer (St. Joseph, MI, USA) [20]. The results were expressed as a percentage of the total sample weight. All measurements were performed in triplicate (*n* = 3) to ensure accuracy and reliability.

### 2.5. GC-MS Analysis of CePs

GC-MS analyses were performed using a Trace GC gas chromatograph equipped with an autosampler (Tri Plus) and a DSQ mass spectrometer quadrupole, all sourced from Thermo Scientific (Waltham, MA, USA). The analytical separation was achieved using a ZB-5 Zebron column from Phenomenex (Torrance, CA, USA), characterized by a stationary phase composition of 5% phenyl and 95% dimethylpolysiloxane, with dimensions of 30 m × 0.25 mm inner diameter (i.d.) × 0.25 μm film thickness. The temperature program for the column began at 80 °C, was maintained for 2 min, was followed by a gradient increase of 5 °C/min, and culminated in a final temperature of 230 °C. Helium was used as the carrier gas at a flow rate of 1.2 mL/min. A sample injection of 1 μL was conducted in splitless mode at an injection temperature of 250 °C. The ion source and mass spectrometer transfer line were maintained at 230 °C. Mass spectral detection was performed under a Select Ion Monitoring (SIM) program, utilizing electron ionization mode (EI) at 70 eV. Trimethylsilyl (TMS) derivatives were identified based on their characteristic retention times and mass spectral fragmentation patterns, which were compared against standards used in monosaccharide identification. Additionally, the compounds were further confirmed by comparing their mass spectra with those available in the National Institute of Standards and Technology (NIST) 2014 mass spectral library.

### 2.6. FT-IR Characterisation of CePs

The FT-IR spectra of CePs were recorded using a Shimadzu IRAffinity-1S spectrophotometer equipped with QATR 10 accessory (Shimadzu Corporation, Kyoto, Japan). The Attenuated Total Reflection (ATR) mode for FT-IR spectroscopy, due to its advantages in analyzing solid samples like polysaccharide extracts, allows for minimal sample preparation and high sensitivity, which is crucial for accurately characterizing their chemical composition [21,22]. The instrument included a DLATGS detector and utilized LabSolutions IR software version 2.10 (Shimadzu Corporation, Kyoto, Japan) for data analysis. Spectra were acquired in the 400 to 4000 cm^−1^ range with a resolution of 4 cm^−1^, conducting 60 scans for each spectrum using the potassium bromide (KBr) disc technique.

### 2.7. Determination of Total Sugar Content of CePS

The total sugar content of the CePs was determined using the anthrone test, with Glc serving as a standard [23]. An anthrone solution was initially prepared at a concentration of 2 mg mL^−1^ in H_2_SO_4_. A solution of 10 mg mL^−1^ of CePs in water was then prepared and diluted 100-fold. Glucose dilutions ranging from 200 μg mL^−1^ to 10 μg mL^−1^ in water were utilized to construct the calibration curve. Then, 200 μL of each sample was combined with 1 mL of the anthrone solution. The resulting mixture was hydrolyzed at 100 °C for 15 min. After hydrolysis, 200 μL of each sample was transferred to an SPL 96-Well Plate (Type 35096) for measurement. The absorbance was recorded at 620 nm using a BioTek Epoch™ microplate reader (Agilent Technologies, Santa Clara, CA, USA). All measurements were performed in triplicate to ensure accuracy and reproducibility.

### 2.8. Determination of Total Phenolic Content of CePs

The total phenolic content was measured using the modified Folin–Ciocalteu method [24]. All colorimetric measurements were performed on the SPL 96-Well Plate (Type 35096) using a BioTek Epoch™ microplate reader (Agilent Technologies, Santa Clara, CA, USA). A CePs solution was prepared at 10 mg mL^−1^ in deionized water. A sodium carbonate (Na_2_CO_3_) solution at 200 mg mL^−1^ was also prepared. Once the required reagents were ready, 400 μL of distilled water, 20 μL of the sample, and 40 μL of Folin-Ciocalteu reagent were mixed with a V1 plus vortex (Boeco, Hamburg, Germany) and allowed to stand for 5 min. Next, 200 μL of the Na_2_CO_3_ solution was added and allowed to react for one hour in the dark. After the reaction, 200 μL of each sample was transferred to the microplate, and the absorbance was measured at 765 nm. For the calibration curve, gallic acid was used in a concentration range from 200 μg mL^−1^ to 10 μg mL^−1^ in H_2_O. The results were expressed as milligrams of gallic acid equivalent (mg of GAE) per gram of the extract’s dry weight (DW). All tests were conducted in triplicate.

### 2.9. Determination of Protein Content of CePs

The protein content of the CePs was measured using the Lowry method [25]. For this analysis, 20 mg of CePs were homogenized in 5 mL of 0.1 M sodium hydroxide (NaOH) at 100 °C for 1 h. The extract was centrifuged at 1792× *g* for 15 min using a Sartorius 2-16PK centrifuge (Sigma, Steinheim, Germany). Subsequently, 250 µL of the supernatant was mixed with 750 µL of NaOH. This solution was combined with 1 mL of Lowry reagent and incubated at room temperature (25 °C) for 10 min. After incubation, 100 µL of Folin reagent (Sigma-Aldrich, USA) was added, and the mixture was vortexed immediately. After 30 min in the dark, the absorbance of the sample was measured at a wavelength of 750 nm using a Micro Plate Reader 2001 spectrophotometer (Whittaker Bioproducts, East Rutherford, NJ, USA). A standard curve was established using bovine serum albumin (BSA) (Sigma-Aldrich, St. Louis, MO, USA).

### 2.10. Antioxidant Activity of CePs

#### 2.10.1. DPPH Radical Scavenging Activity of CePs

The DPPH radical-scavenging activity was evaluated according to the method described in previous studies [26]. The solutions used in the antioxidant tests were prepared 24 h prior to each test. 10 mg of CePs was dissolved in 1 mL of distilled water to create the stock solution. The solution was then sonicated for 40 min at 40 °C using Machtig PS-30A ultrasonic equipment to ensure the complete solubilization of the CePs. Serial dilutions were prepared from the stock solution to obtain concentrations ranging from 10,000 to 312.5 μg mL^−1^ for testing.

For the control, 10 mg of ascorbic acid was dissolved in 1 mL of distilled water to create a stock solution, which was diluted in the same manner as the CePs. The EC_50_ value (mg mL^−1^) was defined as the effective concentration at which 50% of the DPPH radicals were scavenged. All tests were performed in triplicate to ensure reliability. The scavenging ability of DPPH radicals was calculated using the formula:Scavenging ability (%) = (1 − As/Ac) × 100
where (Ac) represents the absorbance of control without CePs, and (As) is the absorbance in the presence of the CePs.

#### 2.10.2. ABTS Radical Scavenging Activity of CePs

The ABTS^•+^ assay was conducted with modifications following the procedure outlined in previous studies from [27]. A solution of 50 mL of 7 mM ABTS (2,2-azino-bis-3-ethylbenzothiazoline-6-sulfonate) and 2.45 mM potassium persulfate (K_2_S_2_O_8_) was prepared and incubated in the dark for 12–16 h to facilitate the generation of the ABTS^•+^ radical cation. The two stock solutions were mixed in equal parts and allowed to stand in the dark for an additional 16 h at room temperature to ensure complete generation of ABTS.

The concentrated ABTS^•+^ solution was then diluted in a 1:14 ratio (ABTS^•+^/distilled water) to achieve an absorbance of 0.70 ± 0.01 units at 734 nm, as measured using a BioTek Epoch™ microplate reader. Subsequently, 20 µL of the sample and 180 µL of the ABTS^•+^ radical solution were added to an SPL 96-Well Plate (Type 35096) and incubated for 30 min. Absorbance was measured again at 734 nm using the BioTek Epoch™ microplate reader. All measurements were performed in triplicate to ensure reliability and accuracy.

The antioxidant activity was expressed as percentage inhibition, calculated using the formula:ABTS radical scavenging capacity [%] = [(Ac − As)/Ac] × 100
where (Ac) represents the absorbance of the control and (As) represents the absorbance of the sample.

#### 2.10.3. Total Antioxidant Capacity (TEAC)

A 1 mg mL^−1^ Trolox stock solution, a water-soluble vitamin E analog, was prepared in EtOH to establish the TEAC curves [28]. This stock prepared the concentrations used for the ABTS curve (100, 80, 60, 40, and 20 μg mL^−1^) and DPPH (10, 8, 6, 4, and 2 μg mL^−1^). The Trolox standard curve was generated by plotting the percentage inhibition against Trolox concentration. The results of the assays were expressed as micromoles of Trolox equivalents per gram (µmol TEAC/G). All experiments were conducted in triplicate, and values were expressed as averages ± standard deviation.

### 2.11. Biological Assays

#### 2.11.1. PC12 Cells

The PC12 cell line, which is derived from chromaffin cells with ectodermal lineage, was obtained from ATCC (Manassas, VA, USA) and cultured in Dulbecco’s Modified Eagle Medium (DMEM) supplemented with 5% fetal bovine serum (FBS), 5% horse serum, and antibiotics (1% penicillin and streptomycin). The cells were maintained in a thermoregulated environment at 37 °C with 5% CO_2_ until they reached approximately 80% confluence, which is considered optimal for subsequent experimental treatments.

#### 2.11.2. Cytotoxicity Assay (MTT)

Different concentrations of CePs were prepared (1 μg mL^−1^, 10 μg mL^−1^, 25 μg mL^−1^, 50 μg mL^−1^, and 100 μg mL^−1^), along with a positive cytotoxicity control, carbonylcyanide-p-(trifluoromethoxy) phenylhydrazone (FCCP, 1 μM), using phosphate-buffered saline (PBS). PC12 cells were treated with various concentrations of CePs and FCCP (1 μM) for 24 h. Following treatment, the culture medium was removed, and 100 µL of (3-(4,5-dimethylthiazol-2-yl)-2,5-diphenyltetrazolium bromide) MTT solution (1 mg mL^−1^ in 1X DPBS; Sigma-Aldrich, St. Louis, MO, USA) was added to each well. The cells were incubated for 40 min at 37 °C to allow for the formation of formazan crystals, which indicate viable cells. After incubation, the formazan crystals were solubilized with 100 µL of isopropanol, followed by an additional 25-min incubation. Absorbance was measured at 570 nm using a NOVOstar Multiplate Reader (BMG Labtech, Offenburg, Germany). This assay quantifies cell viability based on mitochondrial activity, as viable cells reduce MTT to formazan, while non-viable cells do not. The results were analyzed to assess the cytotoxic effects of CePs on PC12 cells, with a focus on concentration-dependent responses.

#### 2.11.3. Aggregation of Soluble Oligomers of the Aβ1–40 Peptide

The Aβ_1–40_ peptide (GenicBio, Shanghai, China) was reconstituted in dimethyl sulfoxide (DMSO) to achieve a concentration of 2.3 mM. A 2 μL aliquot of the stock solution was diluted in sterile phosphate-buffered saline (PBS; Corning, Manassas, VA, USA) to achieve a final concentration of 80 μM. This solution was then subjected to vertical agitation for 4 h, consisting of 2 h at 37 °C and 2 h at room temperature.

#### 2.11.4. Neuroprotection Assay Against Aβ-Generated Toxicity (MTT)

PC12 cells were treated for 24 h with amyloid-beta (Aβ) at a concentration of 1 µM, FCCP at 1 µM, and a combination of Aβ and CePs at the same concentrations previously employed. The procedure for the neuroprotection assay was identical to that used in the cytotoxicity assay. Specifically, the culture medium was removed post-treatment, and the MTT assay was performed to assess cell viability.

### 2.12. Zebrafish Embryo Toxicity Assay

A variant of the zebrafish embryo acute toxicity assay (ZEFT) was performed as outlined by [29], following the Organization for Economic Cooperation and Development (OECD) Test Guideline No. 236 [30]. This method has been validated under the International Conference on Harmonization (ICH) S5(R3) Guideline (2020) for its high sensitivity and predictability [31]. *Danio rerio* embryos were obtained from matings of AB wild-type adults raised and cultured at the Center of Experimentation and Animal Behavior of the University of Malaga (CECA-UMA) and the Institute of Biomedical Research of Malaga and Nanomedicine Platform (IBIMA-BIONAND platform), using eggs purchased from the European Zebrafish Resource Center (EZRC). The embryos’ husbandry was conducted per the European Directive 2010/63/EU and Spanish Royal Decree 118/2021 (REGA: ES 29 067 0001998).

Four hours post-fertilization (hpf), eggs were placed one per well in a 96-well microplate, with each well containing 300 µL of the tested concentrations. Based on dilutions of CePs in E3 embryo medium, these concentrations ranged from 0.05 to 5 mg mL^−1^ and were derived from a 5 mg mL^−1^ stock solution [32]. Negative controls consisted of E3 embryo medium diluent, while positive controls utilized a 2 mg mL^−1^ ulvan polysaccharide dilution in E3 medium [33]. The experiments were conducted in triplicate over 3 days.

Daily observations of morphological and physiological characteristics (e.g., viability, chorion lysis, cardiac edema, and hatching) were recorded using a magnifying microscope (Nikon SMZ-445) (Nikon Europe B.V., Amstelveen, The Netherlands). Stunted growth was identified when the morphology of embryos did not align with that depicted in the international atlas by [34] for embryos of the same developmental stage (hours post-fertilization). Digital images were captured with a Nikon Microphot-FX Fluorescence Microscope (Nikon DS-L1 camera) (Nikon Europe B.V., Amstelveen, The Netherlands) or a Cytosmart Lux3 microscope (Axion Biosystems, Atlanta, GA, USA). Lethal endpoints were identified based on the presence of embryo coagulation or the absence of a heartbeat. The concentration that resulted in a 50% lethality rate of the tested embryos (50% lethal concentration, LC_50_) was determined according to [35]. The surviving embryos at 120 hpf were euthanized through over-anesthesia.

### 2.13. Statistical Analysis

The results from all three in vitro experiments were expressed as mean ± standard error (*n* = 3), while results from in vivo experiments were presented as mean ± standard deviation (*n* = 3). Before conducting statistical analyses, the assumptions of normality and homogeneity of variance were assessed. A one-way analysis of variance (ANOVA) was performed when these assumptions were met; otherwise, the non-parametric Kruskal-Wallis test was applied to compare differences between groups. In instances where significant differences were identified, post-hoc comparisons were conducted using Tukey’s test for ANOVA or Dunn’s test for the Kruskal-Wallis test, with significance set at *p* ≤ 0.05. All statistical analyses were performed using GraphPad Prism 8 software (GraphPad Software, San Diego, CA, USA), whereas in vivo analyses utilized Statgraphics Centurion 19 (Statgraphics Technology, Inc., The Plains, VA, USA). Mortality-sinking correlations were estimated using Yates’ Chi-Square tests conducted in an Excel spreadsheet (Microsoft).

## 3. Results

### 3.1. Collection and Authentication of Ascocarps

Identifying ascocarps is crucial in research involving fungi such as *C. espinosae* (Figure 1). Accurate species identification is essential for the validity of the study and involves a thorough examination of macro- and micromorphological features [3]. The identity of the ascocarp of *C. espinosae* used in this research was confirmed based on macro and micromorphological features [36]. *C. espinosae* specimens were characterized by their mature stroma, which ranged from 1.5 to 5 cm in diameter and exhibited a fleshy, globose form derived from spherical swellings on *Nothofagus* species. The stroma transitioned from a cream-orange hue in youth to an orange-ochre shade at maturity, with a hollow interior [37]. Its numerous apothecia measured 4 to 6 mm in depth and 3 to 5 mm in diameter, presenting a prism-like to pyramidal shape adorned with a velvety texture and an orange hymenium.

Microscopically, the fungus features eight-spored, cylindrical asci; multilocular paraphyses; and subglobose ascospores measuring 11–14 μm, with thin walls and abundant guttules, displaying an ochre to smoky coloration. *C. espinosae* could be differentiated from *C. berteroi* by its globose shape and smaller size. Additionally, it had a softer consistency compared to *C. hariotii* and *C. darwinii*.

### 3.2. Polysaccharides Characterization

#### 3.2.1. Yield of Polysaccharides

From a total of 74.41 ± 1.97 g of lyophilized fruiting body of *C. espinosae*, we successfully isolated 6.48 ± 0.62 g of CePs, representing an extraction yield of 8.71 ± 0.77% (see Table 1). This yield is consistent with typical extraction rates observed in various fungal species.

#### 3.2.2. Elemental Composition Analysis

Table 2 presents the experimental values for total carbon (TC), total nitrogen (TN), total hydrogen (TH), and total sulfur (TS) content of CePs. These parameters are crucial for characterizing the chemical composition of CePs and offer insights into their structural characteristics and potential functionalities. The results will enhance understanding of the biochemical properties of CePs and their implications for future research and applications.

According to the obtained molar (C/N ratio) value, it was 65.03 in polysaccharides

#### 3.2.3. Monomeric Composition

Monosaccharides extracted from CePs were analyzed using GC-MS. Figure 2 displays distinct peaks for various monosaccharides, while Table 3 summarizes the retention times, isomers, peak areas, and their percentage contributions to the total mass.

#### 3.2.4. FT−IR Analysis

The FT−IR analysis of CePs revealed key features of carbohydrates (Figure 3). A broad band at 3287 cm^−1^ indicated hydroxyl groups, and a weak band at 2921 cm^−1^ suggested C-H stretching. Bands at 1645 and 1531 cm^−1^ were linked to C=O stretching, while bands at 1416 and 1352 cm^−1^ indicated O-H bending. A notable band at 999 cm^−1^ suggested glycosidic linkages, and the bands at 927 and 848 cm^−1^ indicated β and α configurations, respectively. These findings align with existing FT−IR profiles of polysaccharides and carbohydrates [38,39,40].

#### 3.2.5. Total Sugar Content

The total sugar content of the CePs was 76.86 μg mL^−1^, which accounted for 77.36 g/100 g of the CePs (Table 1). This determination was achieved by constructing a calibration curve using Glc as a standard (Appendix A).

#### 3.2.6. Total Protein Content

The total protein content of the CePs was 1.57 ± 0.76 (expressed as g of protein per 100 g of CePs) (Table 1). This finding provides valuable insight into the protein composition of the CePs, indicating the presence of proteins in the sample.

### 3.3. Antioxidant Assays

#### 3.3.1. Radical Neutralization Assay (DPPH)

The DPPH assay was conducted to evaluate the radical neutralization activity of different concentrations of CePs. The percentage inhibition varied significantly (*p* < 0.05), with the highest inhibition of 32.14 ± 1.94% at 10 mg mL^−1^. Other concentrations yielded inhibitions of 24.09 ± 0.81% (5 mg mL^−1^), 14.56 ± 1.67% (2.5 mg mL^−1^), 12.08 ± 1.18% (1.25 mg mL^−1^), 7.91 ± 0.84% (625 μg mL^−1^), and 5.32 ± 1.54% (312.5 μg mL^−1^), demonstrating a concentration-dependent trend (Figure 4). In contrast, ascorbic acid, as a control, showed a significantly higher inhibition of 94.77 ± 0.49%. The IC_50_ for CePs was 16,194 μg mL^−1^, and for Trolox, it was 0.00021 μmol mL^−1^. These values were used to calculate the TEAC of 0.0013 μmol TEAC g^−1^ DW.

#### 3.3.2. Radical Neutralization Test (ABTS^•+^)

The percentage inhibition of ABTS^•+^ by CePs displayed a concentration-dependent trend with significant differences between the tested concentrations (*p* < 0.05) (Figure 4). The highest inhibition value, achieved at a concentration of 10 mg mL^−1^, was 19.10% ± 1.00%. The inhibition percentages for other concentrations were as follows: 10.61% ± 0.26% (5 mg mL^−1^), 6.19% ± 0.26% (2.5 mg mL^−1^), 4.59% ± 0.38% (1.25 mg mL^−1^), 2.40% ± 0.46% (625 μg mL^−1^), and 1.80% ± 0.33% (312.5 μg mL^−1^) (Figure 4). In comparison, ascorbic acid exhibited a significantly higher percentage inhibition than CePs, with a value of 100.1% ± 0.19% across all concentrations tested. Additionally, the IC_50_ of CePs was 26,762 μg mL^−1^, and the IC_50_ of Trolox was 0.0024 μmol mL^−1^. These values were used to calculate the TEAC value, which was determined to be 0.09 μmol TEAC g^−1^ DW.

#### 3.3.3. Total Phenolic Content

The total phenolic content (TPC) in CePs was assessed using the Folin-Ciocalteu colorimetric method. The measured TPC was 0.801 ± 0.34 mg GAE g^−1^ DW (gallic acid equivalents per gram of dry weight) [41].

### 3.4. Neuroprotection Assays

#### 3.4.1. Cell Viability Assay

In this study, FCCP (carbonyl cyanide p-(trifluoromethoxy) phenylhydrazone) was used as a positive control for cell toxicity, resulting in a significant reduction in cell viability of approximately 41.32% (Figure 5). PC12 cells treated with different concentrations of CePs exhibited cell viability values comparable to the control group: 103.0 ± 10.03% (0.1 μg mL^−1^), 109.0 ± 8.49% (1 μg mL^−1^), 123.9 ± 13.25% (10 μg mL^−1^), and 90.04 ± 10.94% (100 μg mL^−1^). These findings suggest that CePs do not induce cytotoxic effects at the tested concentrations.

#### 3.4.2. Cell Neuroprotective Assay

The neuroprotective potential of CePs was evaluated by co-incubating PC12 cells with β-amyloid peptide (Aβ) at a concentration of 1 µM alongside varying concentrations of CePs. The results indicated that Aβ significantly reduced cell viability to 62.68 ± 3.36%, while the cell toxicity control, FCCP, caused a similar reduction of 63.95 ± 2.37%. Importantly, cells treated with Aβ in combination with CePs exhibited a significant recovery in cell viability compared to those incubated solely with Aβ (*p* < 0.05). The cell viability values for different concentrations of CePs were as follows: 75.21 ± 2.11% at 100 μg mL^−1^, 78.54 ± 2.17% at 50 μg mL^−1^, 76.65 ± 2.31% at 25 μg mL^−1^, 75.84 ± 3.49% at 10 μg mL^−1^, and 77.06 ± 3.37% at 1 μg mL^−1^ (Figure 6). These findings suggest that CePs protect against Aβ-induced toxicity in PC12 cells, highlighting their potential as a neuroprotective agent in the context of Alzheimer’s disease.

### 3.5. In Vivo Zebrafish Toxicity Assay

In our study, we employed a zebrafish embryo toxicity (ZFET) assay to assess the toxicity of CePs across a range of concentrations. The LC_50_ of CePs at 72 h post-fertilization (hpf) was determined to be 1.472 ± 0.1 mg mL^−1^, excluding data from the 5 mg mL^−1^ concentration. Zebrafish embryos treated with 5 mg mL^−1^ exhibited no or minimal indications of toxicity at 24, 48, or 72 hpf, when compared to 0 mg mL^−1^ (E3) treated-embryos (Figure 7A,B,E,F,I,J). However, other concentrations resulted in various signs of toxicity. Additionally, zebrafish larvae treated with concentrations below 2 mg mL^−1^ (Figure 7C,G,K) displayed stunted growth like that observed in larvae treated with 5 mg mL^−1^ CePs (Figure 7A,E,I) and the control group. The developmental stage of larvae treated with 5 mg mL^−1^ CePs was comparable to that of the negative control (Compare Figure 7A,B,E,F,I,J). In contrast, larvae treated with 2 or 1 mg mL^−1^ CePs exhibited stunted growth compared to the control groups (Compare Figure 7C,D,G,H,K,L).

The viability results are graphed in Figure 8. At CePs concentrations below 2 mg mL^−1^, the viability percentage was almost 100% during the three days of the experiment, whereas it significantly dropped at 2 mg mL^−1^ concentration at 48 and 72 hpf. Nevertheless, this effect was reduced to 5 mg mL^−1^. At 24 hpf, no mortality was observed; however, mortality increased at 48 hpf, and this effect was statistically significant at 72 hpf.

The embryos did not hatch in any experimental group within 24 h of treatment. However, there was a gradual increase in the percentage of hatched embryos at 48 and 72 h post-fertilization (hpf) in the negative control and at all substance concentrations except for 1 and 2 mg mL^−1^. At these concentrations, there was a non-statistically significant decrease in the average hatching rate, accompanied by an increase in variability. This trend was reversed at 5 mg mL^−1^, where the hatching rate returned to a level observed in the control group or at very low concentrations (Figure 9).

After administering CePs at various dilutions, gel formation was observed within minutes. Despite the significant reduction or absence of gelation at lower concentrations (1 and 2 mg mL^−1^), preliminary indications of gelling were noted at these levels. Conversely, CePs at 5 mg mL^−1^ exhibited complete gelation. Indeed, to achieve the requisite concentrations and distribute them promptly in the 96-well plates, all the necessary steps were completed ad hoc. Subsequently, a single 4-h post-fertilization (hpf) zebrafish embryo was carefully released onto the surface of each solution using fine forceps. The partial gelation did not impede embryo sinking in the lower concentrations; however, minimal sinking was observed at the fully gelled 5 mg mL^−1^ concentration, with embryos remaining near the surface throughout the three-day treatment period. While still within their chorion, the embryos displayed considerable mobility; however, following hatching, the larvae exhibited limited movement both on the surface and within the gel matrix. Observations were conducted using two different microscopy techniques: a conventional microscope for the 5 mg mL^−1^ concentration and an inverted microscope for the lower concentrations and control samples (Figure 8). The gelling behavior of CePs at this concentration may elucidate the apparent contradictions observed in the phenotypic study. Consequently, it can be surmised that the chemical composition of CePs is not toxic to zebrafish embryos at concentrations up to 5 mg mL^−1^.

Nevertheless, the gel/sol state of the solution could impede oxygen diffusion, potentially leading to hypoxia, mortality, growth stunting, and delayed hatching in the embryos. To explore this hypothesis, we correlated the reduced viability observed in 72 hpf larvae treated with 5 mg mL^−1^ CePs (Figure 9) with the positioning of the larvae’s heads over the gel, as recorded in daily digital images. During the emergence process from the chorion, a subset of larvae partially submerged their heads within the surrounding solution. Thus, two conditions were analyzed: those in which the heads sank versus those that did not. We found a significant association between larval viability and the non-sinking of the heads, as determined by the Yates’ Chi-Square test (*p* < 0.05; *n* = 23).

## 4. Discussion

The neuroprotective potential of fungal polysaccharides has been documented in the literature. In this study, the CePs mitigated toxicity induced by the Aβ peptide, likely due to its distinct chemical composition. However, the specific mechanisms behind the bioactivity of these compounds require further research. The observed antioxidant activity of CePs is mainly attributed to their protein content. Although these findings are promising, additional tests are necessary to evaluate the antioxidant mechanisms of polysaccharides more thoroughly.

The high polysaccharide content in fungi is crucial for their structural integrity and signaling functions [42,43]. The main polysaccharides in fungal cell walls are α- and β-glucans [42]. α-glucans are amorphous, water-soluble, and serve as energy reserves, with glycogen being a common form, while other glycans are linked by α-1,3 and α-1,4 glycosidic bonds [44,45,46]. In contrast, β-glucans, which are more abundant, show significant structural variability, with up to seven types identified [39]. These sugars, particularly β-glucans, are noted for their immunomodulatory effects, activating the immune system in animals and plants, influenced by their molecular structure and size [46,47,48].

Polysaccharides’ molecular weights are linked to their biological activity. Research indicates that lower molecular weight polysaccharides (around 465.65 kDa) have stronger antioxidant activities than higher molecular weight ones (approximately 703.45 kDa) [49]. A study showed that the molecular weight of polysaccharides from *Chlorella ellipsoidea* significantly affects their immune activity in RAW264.7 cells [50]. This underscores the need to identify structural characteristics that enhance polysaccharide bioactivity.

Integrating these findings on the extract’s characteristics can enhance our understanding of fungal polysaccharides’ therapeutic applications, particularly in neuroprotection and antioxidant mechanisms. Future research should explore their role in the neurotoxic mechanisms of the amyloid beta peptide to understand their pharmacological actions better and promote their use as a functional food.

### 4.1. Collection and Authentication of Ascocarps

Accurate identification of *C. espinosae* is essential for investigating its potential antioxidant and neuroprotective effects. Misidentification can lead to misleading results regarding its bioactive properties, impacting the validity of health claims associated with the fungi. Phylogenetic studies underline the need for precise taxonomic identification to avoid confusion with morphologically similar species [51]. The antioxidant capabilities of CePs may combat oxidative stress linked to neurodegenerative diseases [52]. Accurate species identification ensures the authenticity of research findings and supports future applications of *C. espinosae* in cognitive health-enhancing functional foods [53].

Understanding the distribution and fruiting characteristics of *C. espinosae* is also vital. This species predominantly grows in temperate forests of southern Chile, favoring specific host trees like *Nothofagus* species (e.g., *N. macrocarpa* and *N. glauca*) and typically fruits from September to November, aligning with the phenological cycles of its hosts [54].

*C. espinosae* is characterized by a distinctive globose shape and a diameter of 1.5 to 5 cm. While, like *C. berteroi*, it is generally larger. The color transitions from cream-orange in its juvenile phase to orange-ochre at maturity, in contrast to the paler hues of *C. darwinii* and *C. hariotii*, which are typically white or light yellow. The surface of *C. espinosae* is smooth and fleshy, whereas *C. hookeri* may be furrowed or scaly. Microscopically, it features eight-spored, cylindrical asci and subglobose ascospores measuring 11–14 μm in diameter, with multi-septate and bifurcated paraphyses forming loose palisade formations. These structural traits help distinguish *C. espinosae* from closely related species, highlighting the significance of morphological and microscopic evaluations. Accurate identification thus supports research authenticity and future applications in cognitive health-enhancing functional foods [55].

### 4.2. Polysaccharides Characterization/Chemical Assessment

#### 4.2.1. Yield of Isolation of CePs

In this study, we obtained 6.48 g of CePs from 74.41 g of lyophilized fungal material, yielding 8.71% of the initial weight. This significant yield suggests the presence of bioactive compounds, similar to those in *H. erinaceus*, known for their neuroprotective effects against oxidative stress-related neuronal damage [56].

Optimized extraction techniques are essential for enhancing the bioactive potential of polysaccharides [57]. Further characterization of CePs is needed to clarify their structures and activities, particularly regarding neuroprotection and antioxidant effects in chronic neurodegenerative diseases [58]. The yield of CePs lays a solid foundation for future therapeutic research in neuroprotection.

#### 4.2.2. Total Carbon (C), Hydrogen (H), Nitrogen (N) and Sulphur (S)

The elemental analysis of CePs revealed a total carbon (TC) content of 37.85%, total hydrogen (TH) at 6.593%, total nitrogen (TN) at 0.679%, and no detectable sulfur (TS). This composition highlights the carbohydrate-rich nature of the polysaccharides, typical of those from fungal sources [59].

The low nitrogen content and absence of sulfur suggest that these polysaccharides mainly consist of neutral sugars, with minimal amino acids or sulfated groups, which are often associated with enhanced biological activities [60]. The nitrogen present may indicate some incorporation of amino sugars or related compounds that could affect the functional properties of the polysaccharides [61].

Additionally, the lack of sulfur points to a deficiency in sulfated polysaccharides, known for their various bioactive properties, such as anticoagulant, immunomodulatory, and antioxidant effects. This elemental composition may limit the biological activities of these CePs. Overall, the analysis sheds light on the structural characteristics and potential applications of CePs, paving the way for further investigations into their functional properties and biological effects.

#### 4.2.3. Analysis of Monosaccharide Composition

The analysis of monosaccharide content in CePs revealed Glc as the predominant component, making up about 99.24% of total monosaccharides. This contrasts with other fungal species, particularly in the *Cordyceps* and *Hericium* genera, which show more diverse monosaccharide profiles. For example, exopolysaccharides from various *Cordyceps* species often include Glc, Man, and Gal in different ratios, with one report describing a ratio of 23:1:2.6 for *Cordyceps sinensis* [62].

*H. erinaceus*, known for its neuroprotective properties, features a unique monosaccharide composition that includes Glc and Gal, contributing to its biological activities like nerve growth stimulation and antioxidant effects [63]. The high Glc content in CePs suggests potential therapeutic applications; however, the limited variety in its monosaccharide composition compared to *Cordyceps* and *Hericium* may restrict its utility in specific nutritional or pharmacological contexts.

Notably, CePs contain only 0.90% galactose, significantly less than species like *Aspergillus fumigatus*, which has higher levels [64]. This low Gal content may impact the functional properties of CePs, as higher Gal levels are associated with enhanced immunomodulatory effects and better gut health through microbiome interactions [58]. In contrast, *Cyttaria* and *Hericium*, with their varied monosaccharide compositions, may exhibit improved biological efficacy in health applications [55,65].

These findings emphasize the need for further research on the polysaccharide profiles of CePs and their health implications, including their interactions within biological systems and potential roles in oxidative stress and cognitive function. Extraction methods like GC-MS are valuable for investigating monosaccharide composition and the health benefits of fungal sources [66].

In summary, while the high Glc content in CePs presents promising health applications, the lower galactose level and lack of diverse monosaccharide species warrant further exploration into the functional attributes of this fungus, particularly when compared to the advantageous compositions found in *Cordyceps* and *Hericium*.

#### 4.2.4. FT-IR Spectra Analysis

FT-IR is a crucial technique for identifying molecular constituents through vibrational transitions. In this study, FT-IR was employed to analyze the chemical composition of CePs, focusing on carbohydrate components.

The spectral data revealed functional groups associated with carbohydrates. A broad absorption band at 3287 cm^−1^ indicated O-H stretching vibrations, characteristic of hydroxyl groups in carbohydrates, suggesting strong hydrogen bonding networks typical of polysaccharides. A weaker band at 2921 cm^−1^ was linked to C-H stretching vibrations of aliphatic hydrocarbons, pointing to the presence of methyl or methylene groups in the carbohydrate structure, indicating polysaccharide complexity [67].

Additionally, bands at 1645 cm^−1^ and 1531 cm^−1^ were related to C=O stretching vibrations of carbonyl groups in reducing sugars [68]. The O-H bending vibrations were confirmed by bands at 1416 cm^−1^ and 1352 cm^−1^, suggesting hydroxyl functionalities and water molecules’ role in carbohydrates [69]. A notable band at 1142 cm^−1^, corresponding to C-O stretching, indicates ether and alcohol functionalities inherent to carbohydrates [68].

The band around 999 cm^−1^ reflects a coupling phenomenon involving C-O-C stretching, C-C stretching, and C-O-H bending modes in monosaccharides, indicative of sugar structural dynamics. Bands at 927 cm^−1^ and 848 cm^−1^ suggest β and α configurations, respectively, indicating the orientation of the anomeric carbon in glycosidic bonds, crucial for understanding carbohydrate structures in CePs [69].

The FT-IR analysis identifies a carbohydrate profile in CePs featuring hydroxyl, carbonyl, and ether functional groups. The detection of glycosidic linkages and the determination of α and β configurations highlight the significance of carbohydrates in the biochemical composition of CePs, supporting established patterns in the literature and enhancing insights into their potential biological activities.

#### 4.2.5. Analysis of Total Sugar Content

The total sugar content in CePs was quantified at 77.36 ± 6.129 g/100 g, indicating a high polysaccharide content. This aligns with studies showing that higher Glc levels in polysaccharides correlate with enhanced antioxidant activity [15]. Specifically, polysaccharides with greater Glc concentrations demonstrate more potent antioxidant properties, highlighting the importance of structural composition in bioactivity [51]. Although molecular weights were not measured in this study, literature indicates that low molecular weight polysaccharides have superior free radical scavenging abilities, emphasizing that sugar content and molecular weight are key determinants of functional potential [52].

#### 4.2.6. Analysis of Total Protein Content

The protein concentration of 1.57 ± 0.76 g/100 g found in CePs underscores their nutritional significance and neuroprotective potential. Fungal polysaccharides, such as those from *Ganoderma lucidum* and *Agaricus bisporus*, show antioxidant and immunomodulatory properties due to their protein content, helping to protect against neurodegenerative diseases [70,71]. *Pleurotus ostreatus* also produces bioactive polysaccharides that enhance antioxidant activity [72]. The unique attributes of the *Cyttaria* genus, which include their parasitic relationship with *Nothofagus* trees [73], add context to these findings.

In summary, the protein content of CePs enriches their nutritional profile and supports potential neuroprotective applications, highlighting the need for further research into their bioactivity and therapeutic strategies for neurodegenerative disorders. *G. lucidum*’s high polysaccharide yield, linked to polysaccharide–protein complexes, contributes to its neuroprotective effects [74]. Optimizing extraction techniques may reveal similar interactions in other fungal polysaccharides. Additionally, polysaccharides from *Tremella fuciformis* can protect neurons from glutamate-induced toxicity, indicating their potential in treating neurodegenerative conditions [75]. The proteins within these polysaccharides likely play a crucial role in regulating neuronal survival pathways.

### 4.3. Antioxidant Activity

#### 4.3.1. In Vitro DPPH Assay

The antioxidant properties of CePs were evaluated using the DPPH assay, showing moderate activity. This contrasts with C. militaris polysaccharides, which demonstrate significant DPPH radical scavenging abilities. Their antioxidant efficacy is linked to composition, including molecular weight and monosaccharide types [76]. The free radical scavenging ability of C. militaris polysaccharides highlights their potential for therapeutic applications in oxidative stress-related diseases.

Interestingly, the antioxidant activity of CePs exceeded previous studies, where significant inhibition was only noted at high concentrations (40 mg mL^−1^) for polysaccharides from *Nothophellinus andinopatagonicus* [73]. Variations in inhibition percentages between DPPH and ABTS assays can be attributed to methodological differences, with the TEAC value serving as a useful comparative measure. The DPPH value for CePs was found to be 0.0013 mol TEAC g^−1^ DW, likely influenced by steric hindrance in the DPPH assay [77]. These findings highlight the importance of using multiple assays to assess antioxidant activity across different polysaccharide sources comprehensively.

#### 4.3.2. In Vitro ABTS Assay

The ABTS assay complements DPPH findings, revealing the antioxidant capacity of polysaccharides. Polysaccharides from *H. erinaceus* showed significant ABTS radical scavenging ability, indicating their potential as antioxidant agents in food and medicinal applications. This study recorded a value for CePs at 0.090 mol TEAC g^−1^ DW, significantly higher than the DPPH value of 0.0013 mol TEAC g^−1^ DW, suggesting stronger scavenging activity toward ABTS radicals. CePs are primarily composed of Glc (both isomers) and lack sulfate groups, which may influence the observed antioxidant activity. Variations in activity are likely due to monosaccharide composition and structural characteristics rather than sulfates [59].

### 4.4. Analysis of TEAC Assay

The TEAC assay, when integrated with DPPH and ABTS methods, provides a comprehensive evaluation of antioxidant potential. It measures the ability of compounds to counteract oxidative stress through Trolox equivalent values, allowing for direct comparisons of various antioxidants [78]. This assay is particularly effective in assessing the antioxidant capabilities of polysaccharides from mushrooms, such as *Hericium* and *Cordyceps*, which demonstrate significant activity potentially surpassing conventional antioxidants due to their complex structures.

While the focus has been on free radical scavenging capacity, additional research is needed to investigate other antioxidant mechanisms, including regulation of antioxidant systems and oxidative stress signaling pathways [79]. The TEAC results for CePs showed a value of 0.090 mol TEAC g^−1^ DW in the ABTS assay, notably higher than the DPPH value of 0.0013 mol TEAC g^−1^ DW. This discrepancy may be due to methodological variations, particularly steric hindrance affecting the DPPH assay [77]. The ABTS assay appears more sensitive to CePs’ antioxidant activity, emphasizing the necessity of using multiple assays for a well-rounded understanding of antioxidant potentials, as different methods can yield varied results based on their mechanisms.

### 4.5. Analysis of Total Phenolic Content

The study found the total phenolic content of CePs to be 0.801 mg GAE g^−1^ DW. Quantifying phenolic content is important due to its known antioxidant properties and health benefits. The methods of dialysis and reprecipitation likely aimed to isolate and quantify the phenolic compounds effectively. This quantification provides insights into the antioxidant potential of CePs, as these compounds help scavenge free radicals and reduce oxidative stress [80]. The findings support previous research on the significance of phenolic compounds in plant extracts and their correlation with antioxidant activity [81]. Understanding the phenolic content in CePs enhances knowledge of *C. espinosae*’s bioactive components and its potential as a source of natural antioxidants. Future studies on specific phenolic compounds could reveal more about the health benefits of CePs.

### 4.6. Neuroprotective Activity

The neuroprotective potential of polysaccharides in mitigating β-amyloid (Aβ) peptide-induced toxicity is well-documented [56,82,83]. Studies show that polysaccharides from various organisms enhance cell viability in Aβ-exposed cell lines through mechanisms such as modulating amyloid precursor protein (APP) levels, reducing β-secretase activity, and enhancing Aβ degradation via insulin-degrading enzyme (IDE) and neprilysin (NEP). Additionally, they inhibit Aβ aggregation and exhibit anti-apoptotic properties by reducing cytochrome c release, thus decreasing apoptosis mediated by caspase-3 and possibly activating the PI3K-Akt signaling pathway for neuronal survival.

In this study, the neuroprotective effects of CePs against Aβ-induced toxicity were evaluated in PC12 cells. CePs did not exhibit cytotoxicity, maintaining cell viability similar to controls. While Aβ significantly decreased cell viability, co-administration of CePs with Aβ moderately improved cell viability, indicating that CePs exhibit neuroprotective activity against Aβ toxicity, independent of concentration. This aligns with previous findings that various polysaccharides improve cell viability under Aβ exposure [82,83].

Understanding how CePs mitigate Aβ toxicity is essential, particularly regarding their monosaccharide composition. Polysaccharides with neuroprotective properties often contain a notable proportion of Glc; for instance, those from *H. erinaceus* and *Lycium barbarum*, known for their neuroprotective effects, have high Glc content [82]. Polysaccharides with anti-aggregation properties typically comprise Glc, Man, Gal, and Ara, which may be crucial for inhibiting Aβ aggregation and toxicity.

To advance this research, isolating CePs fractions with over 95% sugar content may provide insights into the structural features and compositions that confer neuroprotective potential against Aβ-induced toxicity in PC12 cells.

### 4.7. Toxicity of the Extract

CePs were found to be non-toxic to the PC-12 cell line and zebrafish embryos, with MTT assays showing no cytotoxicity below 100 μg mL^−1^ and zebrafish tests indicating non-toxicity even at 5000 μg mL^−1^. However, the gel/sol state may negatively impact embryo viability and growth by reducing oxygen diffusion. Hypoxia in zebrafish has been linked to delayed growth and alterations in HIF1A-dependent mechanisms [84,85]. Indicators of acute lethality include growth retardation and lack of heartbeat [30]. While some studies report varying susceptibility of early zebrafish embryos to polysaccharides (e.g., non-toxic *Fucus vesiculosus* [86] vs. toxic extracts from other algae between 245 µg mL^−1^ and 5 mg mL^−1^ [86,87]), growth stunting from *Ulva rigida* polysaccharides has been noted as pre-lethal [33]. Variability in effects may stem from hypoxia due to gelling. Maintaining optimal oxygen levels in zebrafish tests is critical as per OECD guidelines (Test No. 236, 2013). Lastly, pigment depletion could suggest potential anticancer activity when toxicity is low or absent; however, this phenotype was not observed in this study [86].

## 5. Conclusions

This study demonstrates that the CePs attenuate Aβ peptide-induced toxicity, likely due to their chemical composition. While the mechanisms underlying the neuroprotective bioactivity of fungal polysaccharides remain unclear, our findings indicate that these extracts are non-toxic to PC-12 cells and zebrafish embryos, exhibiting antioxidant properties primarily linked to their protein content. However, further research is necessary to explore additional antioxidant mechanisms and elucidate the bioactive properties of these polysaccharides. Future studies are essential to deepen our understanding and uncover potential neuroprotective applications.

## Figures and Tables

**Figure 1 jof-11-00637-f001:**
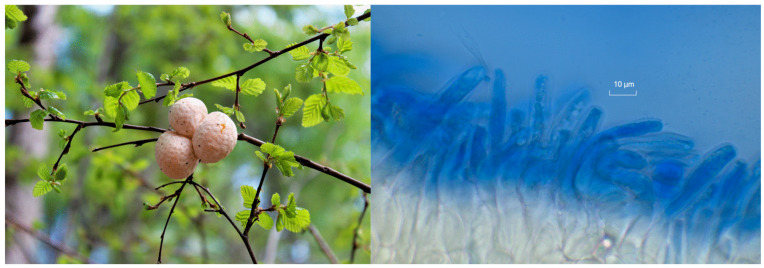
Fruiting bodies and immature asci in lactophenol cotton blue of the *C. espinosae* mushroom on a branch of *N. obliqua*.

**Figure 2 jof-11-00637-f002:**
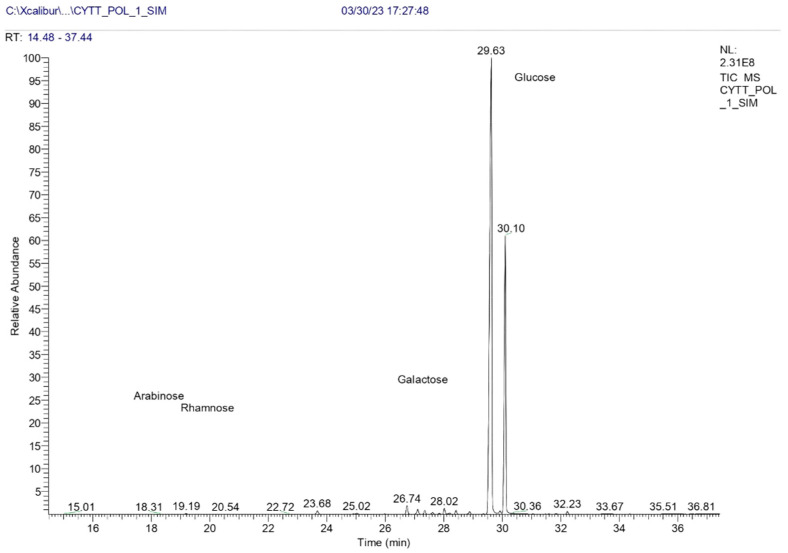
GC−MS analysis of CePs.

**Figure 3 jof-11-00637-f003:**
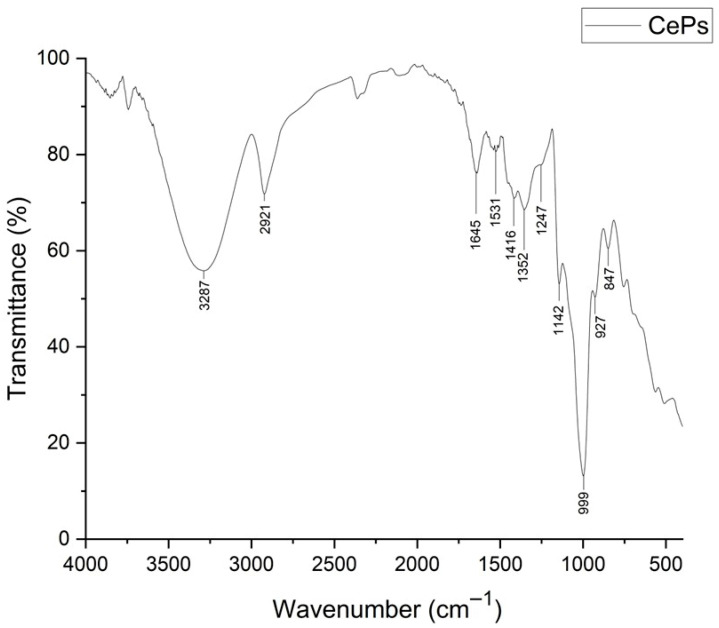
FT−IR spectra of the CePs.

**Figure 4 jof-11-00637-f004:**
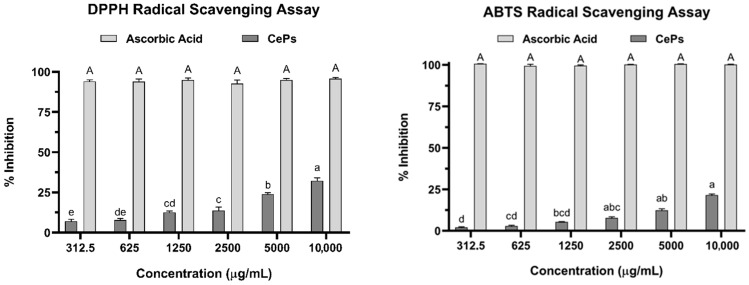
Scavenging effects (%) of CePs on DPPH and ABTS radicals at varying concentrations. The results are expressed as the mean of three replicate measurements, with error bars representing the standard error. Analysis of Variance (ANOVA) was performed, and different letters indicate significant differences between samples at the 0.05 level.

**Figure 5 jof-11-00637-f005:**
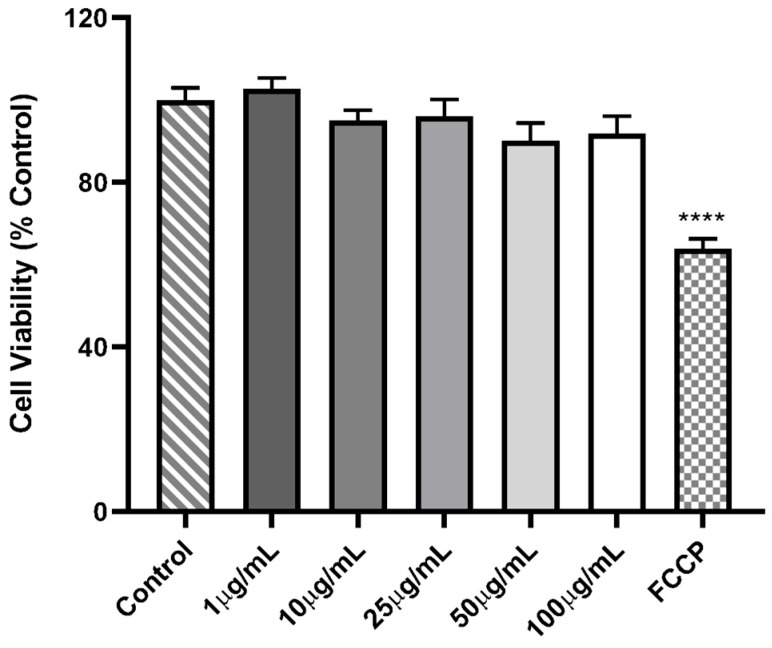
Viability of PC-12 cells incubated with FCCP (1 μM) and different concentrations of CePs (1–100 µg mL^−1^). Data is presented as mean ± standard error from four experiments. The number of asterisks (*) indicates the level of significance of the differences for the control: **** (*p* < 0.0001).

**Figure 6 jof-11-00637-f006:**
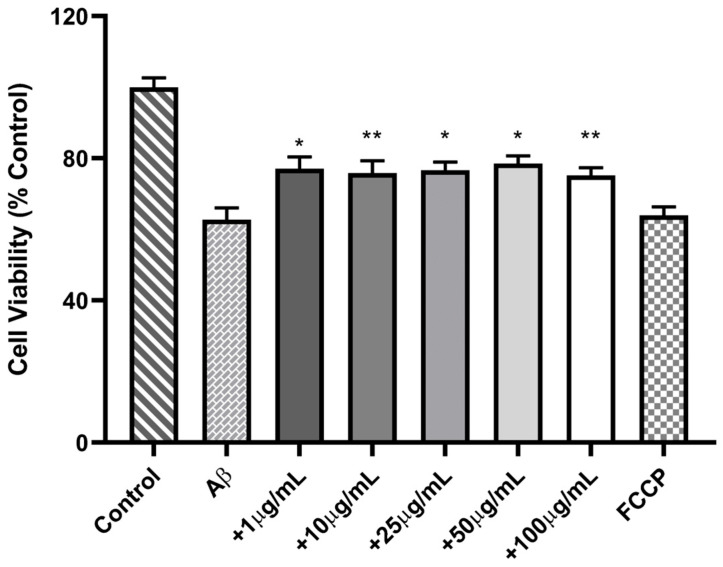
Viability of PC-12 cells co-incubated with different concentrations of CePs (1–100 µgmL^−1^) and β-amyloid (Aβ) at 1 μM. Data are presented as mean ± standard error from four experiments. The number of asterisks (*) indicates the level of significance of the differences compared to the control: * (*p* < 0.05), ** (*p* < 0.01).

**Figure 7 jof-11-00637-f007:**
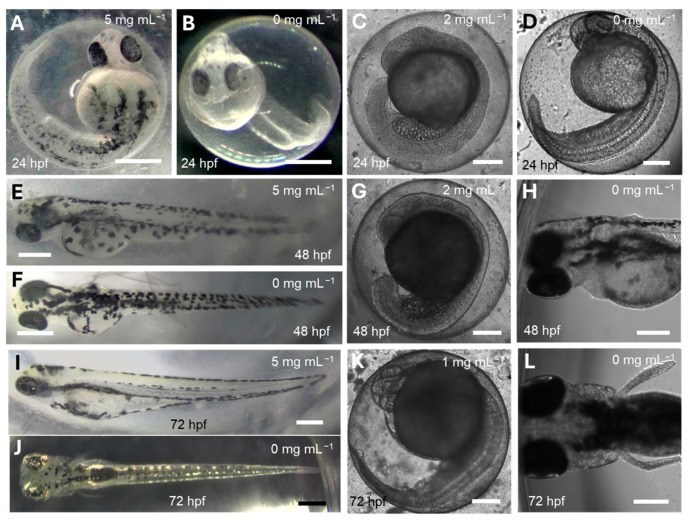
The provided data illustrate the impact of different concentrations of CePs (1–5 mg mL^−1^) on the growth of zebrafish larvae. Panels (**A**–**L**) represent the larvae treated at 24 h (**A**–**D**), 48 h (**E**–**H**), and 72 h (**I**–**L**) post-fertilization (hpf). Zebrafish larvae treated with gelled 5 mg mL^−1^ CePs (**A**,**E**,**I**) exhibited developmental progress closely resembling the control group treated with E3 medium (**B**,**F**,**J**) when allowed to grow on the surface. This similarity was confirmed by Yates’s chi-squared test (*p* < 0.05; *n* = 23). The images marked (**A**,**B**,**E**,**F**,**I**,**J**) were captured using a Nikon Microphot-FX microscope (Nikon Europe B.V., Amstelveen, The Netherlands), while images labeled (**C**,**D**,**G**,**H**,**K**,**L**) were taken with an inverted Cytosmart Lux3 microscope (Axion Biosystems, Atlanta, GA, USA). Bars are 200 (**C**,**D**,**G**,**H**,**K**,**L**) and 500 (**A**,**B**,**E**,**F**,**I**,**J**) μm.

**Figure 8 jof-11-00637-f008:**
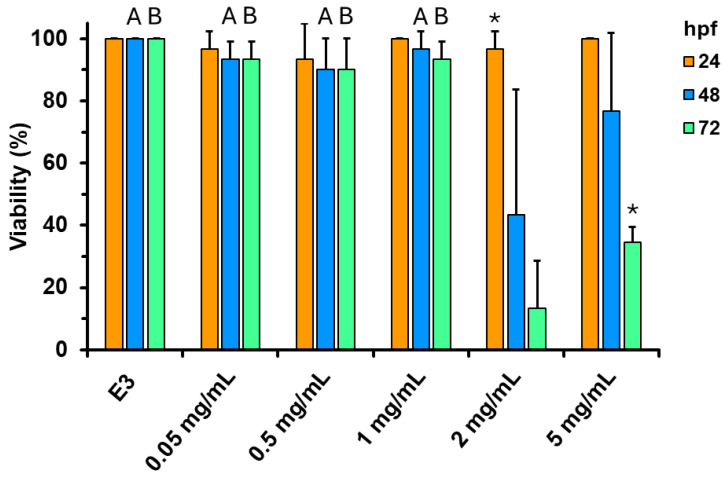
The viability percentage of zebrafish larvae incubated with different concentrations of CePs (0.05–5 mg mL^−1^). Values are means, and vertical bars represent standard deviations calculated from three replicas. A,B: Groups showing statistical difference (*p* < 0.05) with 2 mg mL^−1^ (A) or 2 and 5 mg mL^−1^ (B) values. * Is a statistical difference (*p* < 0.05) with the means of the other two temporal groups of the same concentration. E3 is control zebrafish embryo medium 3 (0 mg/mL CePs).

**Figure 9 jof-11-00637-f009:**
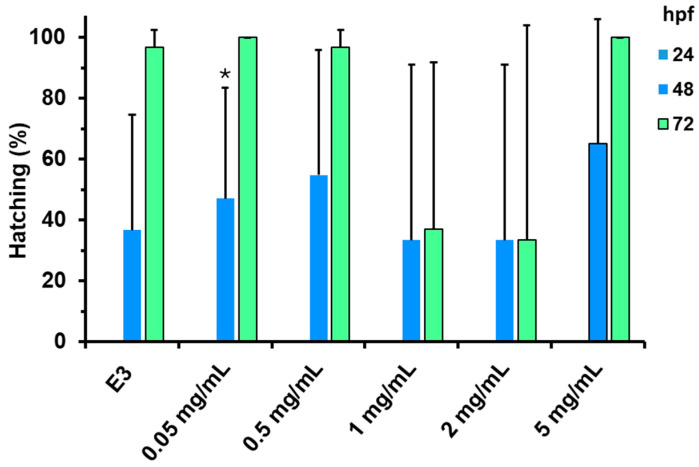
Hatching percentage of zebrafish larvae incubated with different concentrations of CePs (0.05–5 mg mL^−1^). The values presented represent the mean and standard deviation calculated from three replicates. Asterisks (*) indicate statistically significant differences (*p* < 0.05) when compared to the 72 h post-fertilization (hpf) data. All groups, except those at concentrations of 1 mg mL^−1^ and 2 mg mL^−1^, exhibited statistically significant differences compared to the absence of hatching at 24 hpf (*p* < 0.05). E3 is control zebrafish embryo medium 3 (0 mg/mL CePs).

**Table 1 jof-11-00637-t001:** Yield and Composition of Total Phenolics, Total Sugars, and Protein Content.

Yield ^a^ (%)	Total Phenolics (mg GAE g^−1^ DW) ^b^	Total Sugar Content (g/100 g) ^b^	Total Protein (g/100 g) ^b^
8.71 ± 0.77	0.801 ± 0.34	77.36 ± 6129	1.57 ± 0.76

^a^ The value is based on the freeze-dried fruiting body of *C. espinosae*. ^b^ The values are on a dry weight basis of CePs. Data represent mean ± SD (*n* = 3).

**Table 2 jof-11-00637-t002:** Elemental Analysis of CePs.

Elemental Composition of CePs (%)			
TC	TH	TN	TS
37.85 ± 0.5	6.593 ± 0.2	0.679 ± 0.09	0.000 ± 0.002

*n* = 3 (represents the average of three measurements from each point).

**Table 3 jof-11-00637-t003:** Content of monosaccharides in CePs.

Header	Monosaccharide	Retention Time (min)	Peak Area	% Mass
**1**	D (+) Gal isomer 1	28.01	10,354,902	0.60
**2**	D (+) Gal isomer 2	28.88	5,186,177	0.30
**3**	D (+) Glc isomer 1	29.63	1,187,411,268	69.14
**4**	D (+) Glc isomer 2	30.10	512,578,595	29.85
**5**	D (+) Glc isomer 3	32.23	4,288,197	0.25
**6**	D (+) Rha isomer 1	19.18	1,159,481	0.07
**7**	D (+) Rha isomer 2	19.50	238,482	0.01
**8**	D (+) Rha isomer 3	21.83	114,412	0.01
**9**	D (+) Ara isomer 1	18.30	157,850	0.01
**10**	D (+) Ara isomer2	18.64	143,961	0.01

## Data Availability

The original contributions presented in this study are included in the article. Further inquiries can be directed to the corresponding author.

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
