# Peer review of "Potential Antioxidant and Neuroprotective Effect of Polysaccharide Isolated from Digüeñe Cyttaria espinosae"

_jof, 2025, doi:10.3390/jof11090637_

Round 1
Reviewer 1 Report
- You mentioned that Cytaria espinosae used was identified by macro and micromorphological features. Nevertheless, you did not show any additional pictures and microphotographs which confirm your results (lines 363-369, 674-678).
- About the yield of the Isolation of Fruiting Body Polysaccharides, you report 8.71(g/100 g) but also 63.5± 1.2(g/100 g) of polysaccharide content in the extract (also lines 686-689). Please clarify. The same in total protein content of the CePs was 13.6% (expressed as mg of protein per 100 g of CePs).You do not refer that in table 1.
Some minor details for improvements
Use italic letter for the scientific names trough the text.
Please, describe Aβ (as Aβ ,amyloid beta) in the abstract for first time (line 30). Then you can use just “Aβ” trough the text.
Author Response
Reviewer 1
Major comments
- You mentioned that Cytaria espinosae used was identified by macro and micromorphological features. Nevertheless, you did not show any additional pictures and microphotographs which confirm your results (lines 363-369, 674-678).
In response to the insightful comment from Reviewer 1 regarding the omission of photographic evidence for the identification of Cytaria espinosae, we acknowledge the importance of visual documentation in confirming our findings. To address this concern, we will include micrograph that demonstrates the micromorphological features of Cytaria espinosae in the manuscript. This addition will be incorporated into the specified section;
This section incorporates suggestions from reviewers 1 and 2 (In section 2.2.2. provide in detail the method for identifying Cyttaria espinosae samples.). Two references (17 and 18) have been included.
2.2 Collection and Identification of Cyttaria espinosae
2.2.2. Identification
In the identification of Cyttaria espinosae samples, we carefully documented the macroscopic characteristics of the stromata in situ, utilizing photographic documentation via a Canon T6 reflex camera for both field and laboratory observations. This step is vital for establishing a reliable morphological baseline for identification purposes. Taxonomic determination was then achieved through an examination of microscopic features. Stromata morphology was examined using a Motic BA210S Trinocular compound microscope (Barcelona, Spain) with a Moticam S6 camera attached. Microscopic structures were studied by mounting tissue sections of dried stromata in demineralized water and lactophenol cotton blue following Gamundí (1971)[16].
Post-identification, we ensured that the stromata were labeled meticulously and subsequently frozen at -20°C. Following freezing, the stromata underwent lyophilization for 48 hours using a CHRIST Alpha 2-4 LO plus automatic lyophilizer. This process extends the long-term viability of the samples while preventing structural degradation due to ice crystal formation (Villalobos-Pezos et al., 2024) [17]. The final step involved storing the lyophilized specimens at -20°C, preparing them for future analytical research on their nutritional and bioactive properties (Sorokulova et al., 2015) [18].
[17]. Villalobos-Pezos, J.; González, P.; Córdova, E.; Duran, R.; Antezana, M. Effects of Drying Treatments on the Physicochemical Characteristics and Antioxidant Properties of the Edible Wild Mushroom Cyttaria espinosae Lloyd. Journal of Fungi 2024, 10, 80. doi:10.3390/jof11010013.
[18] Sorokulova, I.; Lantukh, D.; Kruchinina, M.; Sabitov, A. Biopolymers for sample collection, protection, and preservation. Applied Microbiology and Biotechnology 2015, 99, 5995–6003. doi:10.1007/s00253-015-6681-3.
3.1 Collection and Authentication of Ascocarps

Figure 1. Fruiting bodies and immature asci in lactophenol cotton blue of the C. espinosae mushroom on a branch of N. obliqua.)
- About the yield of the Isolation of Fruiting Body Polysaccharides, you report 8.71(g/100 g) but also 63.5± 1.2 (g/100 g) of polysaccharide content in the extract (also lines 686-689). Please clarify.
In response to the valuable comment from the reviewer regarding the discrepancies in the yield of fruiting body polysaccharides and total protein content reported in our manuscript, we appreciate your meticulous attention to detail, which is vital in scientific research.
To clarify, the yield of polysaccharides from the isolation of fruiting bodies of Cytaria espinosae was reported as 8.71% of the lyophilized material. This represents the percentage yield of lyophilized polysaccharides obtained post-extraction relative to the lyophilized weight of the fruiting bodies used.
On the other hand, the figure of 63.5 ± 1.2 g per 100 g was removed from Table 1; this was an error.
- The same in total protein content of the CePs was 13.6% (expressed as mg of protein per 100 g of CePs). You do not refer that in Table 1.
Furthermore, regarding the total protein content, we acknowledge the need for clarification in our presentation of results. The reported protein content of 1.57± 0.76 reflects the concentration of protein present expressed as grams of protein per 100 grams of extracted polysaccharides (CePs). This information, however, was inadvertently omitted from Table 1.
To address these inconsistencies, we will revise the manuscript accordingly. Specifically, we will ensure the following adjustments are made:
Table 1. Yield and Composition of Total Phenolics, Total Sugars, and Protein Content in Polysaccharides Extracted from C. espinosa
|
Yielda (%) |
Total phenolics (mg GAE g-1 DW)b |
Total sugar content (g/100 g)b |
Total protein (g/100 g) |
|
8.71 |
0.801±0.34 |
77.36±6,129 |
1.57± 0.76 |
a The value is based on the freeze-dried fruiting body of C. espinosae.
b The values are on a dry weight basis of polysaccharide extract (CePs). Data represent mean ± SD (n = 3).
We believe these amendments will enhance the clarity and accuracy of our findings. Thank you for highlighting these issues, and we are committed to ensuring that our results are presented in a precise and comprehensible manner.
Section 4.2.6 Total protein content
In response to the reviewer’s comments, I have revised section 4.2.6 to enhance the discussion surrounding the total protein content of Cyttaria espinosae (CePs). The updated analysis centers on the neuroprotective potential of the protein concentration, which was measured at 1.57 ± 0.76 g/100 g.
This shift in focus allows for a more comprehensive understanding of the significance of CePs beyond their nutritional profile. Notably, the protein content is linked to potential benefits in mitigating oxidative stress and supporting neuronal health, aligning with findings from various studies on other fungi. Previous literature indicates that polysaccharides derived from fungi can have considerable antioxidant effects, which are crucial in protecting against neurodegenerative conditions.
By emphasizing the integral role of proteins within these polysaccharide complexes, the revised section highlights their possible contributions to cellular signaling pathways and the modulation of neuronal survival. This expanded discussion not only addresses the reviewer's feedback but also opens avenues for future research into the bioactivity of CePs and their therapeutic potential in neurodegenerative diseases.
4.2.6 Total protein content
The protein concentration of 1.57 ± 0.76 g/100 g found in polysaccharides extracted from C. espinosae (CePs) highlights not only their nutritional significance but also their promising neuroprotective potential. This observation gains more weight when compared with findings from other fungal polysaccharide sources.
Studies have consistently shown that mushroom-derived polysaccharides can combat oxidative stress and support neuronal health. For example, polysaccharides from Ganoderma lucidum and Agaricus bisporus have demonstrated robust antioxidant and immunomodulatory properties—effects linked to the proteins integrated within the polysaccharide structures [62].
These proteins are believed to play an essential role in modulating cellular signaling pathways that protect against neurodegenerative conditions. Additionally, polysaccharides from edible fungi are recognized for their strong antioxidant capacities, crucial for reducing oxidative damage in neurons [63]. A notable example is Pleurotus ostreatus, which produces bioactive polysaccharides that have been shown to enhance antioxidant activity and offer potential therapeutic benefits [64].
Research into Ganoderma lucidum revealed that its high polysaccharide yield is associated with specific polysaccharide–protein complexes, which are key contributors to both antioxidant and neuroprotective effects [65]. This insight suggests that by optimizing extraction techniques, similar beneficial interactions may be identified in CePs.
Beyond oxidative stress, fungal polysaccharides have also been shown to modulate apoptosis, a critical process for maintaining neuronal health. Evidence from Tremella fuciformis demonstrates its polysaccharides' ability to shield neurons from glutamate-induced toxicity, reinforcing their potential in treating neurodegenerative diseases [66]. The protein component in these polysaccharides likely plays a role in regulating neuronal survival pathways.
Additionally, understanding the unique attributes of the Cyttaria genus, including their parasitic nature on Nothofagus trees [33], may provide context for these findings and their implications for ecological interactions.
In summary, the protein content of CePs not only enriches their nutritional profile but also lays a foundation for possible neuroprotective applications. These findings encourage further research into the precise mechanisms behind their bioactivity and their role in developing new therapeutic strategies for neurodegenerative disorders.
References added
- Kozarski, M., S.; Klaus, A., S.; Niksic, M., P.; Van, G., Leo; Vrvic, M., M.; Jakovljevic, D., M. Polysaccharides of Higher Fungi: Biological Role, Structure, and Antioxidative Activity. Hem. Ind. 2014, 68, 305–320, doi:10.2298/hemind121114056k.
- Li, H.; Ding, F.; Xiao, L.; Shi, R.; Wang, H.; Han, W.; Huang, Z. Food-Derived Antioxidant Polysaccharides and Their Pharmacological Potential in Neurodegenerative Diseases. Nutrients 2017, 9, doi:10.3390/nu9070778..
- Vamanu, E. Antioxidant Properties of Polysaccharides Obtained by Batch Cultivation of Pleurotus Ostreatus Mycelium. Nat. Prod. Res. 2013, 27, 1115–1118, doi:10.1080/14786419.2012.704376.
- Wang, Q.; Xu, M.; Zhao, L.; Chen, L.; Ding, Z. Novel Insights into the Mechanism Underlying High Polysaccharide Yield in Submerged Culture of Ganoderma lucidum Revealed by Transcriptome and Proteome Analyses. Microorganisms 2023, 11, 772. https://doi.org/10.3390/microorganisms11030772
- Jin, Y.; Hu, X.; Zhang, Y.; Liu, T. Studies on the Purification of Polysaccharides Separated from Tremella Fuciformis and Their Neuroprotective Effect. Mol. Med. Rep. 2016, 13, 3985–3992, doi:10.3892/mmr.2016.5026.
Detailed comments
- Some minor details for improvements
4.1 Use italic letter for the scientific names trough the text.
Thank you for the feedback regarding the use of italics for scientific names. We will ensure that all scientific names, such as Cytaria espinosae, are italicized throughout the manuscript for consistency and clarity.
Line 54 Cordyceps cicadae
Line 58 Cordyceps militaris
Line 61 C. militaris
Line 64 Hericium erinaceus
Line 77 Nothofagus and Cyttaria
Line 84 Cyttaria espinosae
Line 825 Cordyceps militaris
Line 829 C. militaris
Line 862 Hericium and Cordyceps
Line 994 Ulva rigida
Line 955 Cyttaria espinosae
Line 1120 Cyttaria
- Please, describe Aβ(as Aβ ,amyloid beta) in the abstract for first time (line 30). Then you can use just “Aβ” trough the text.
We will consistently refer to amyloid beta as Aβ throughout the manuscript after introducing the full term earlier in the abstract. The revised sentence in line now reads:
“CePs showed no toxicity in zebrafish embryos and maintained high cell viability in PC-12 cells exposed to amyloid β peptide(Aβ).”
Thank you for your valuable suggestions, which have significantly improved the manuscript.
Reviewer 2 Report
The manuscript on the topic "Potential antioxidant and neuroprotective effect of polysaccharide isolated from digüeñe Cyttaria espinosae" refers to experimental articles, is clearly and well-structured. The authors conducted a study of the antioxidant and neuroprotective effects of polysaccharides from Cyttaria espinosae on neurotoxicity induced by beta-amyloid (Aβ) in neuron-like cells. The toxicity of this polysaccharide was assessed in vivo in zebrafish embryos. It was found that CePs have antioxidant and neuroprotective effects and are non-toxic to zebrafish embryos.
The results of this study are the basis for further study of properties in the context of the physiopathology of Alzheimer's disease.
There are several minor comments on the article:
- In section 2.2.2. provide in detail the method for identifying Cyttaria espinosae samples.
- Sections 3.2.1 and 3.3.3 need to be expanded. Provide the calibration curve by which the total sugar content in CeP was determined.
- In Figure 4, it is advisable to indicate the control (ascorbic acid) and provide explanations for A, B, C, D, E in the figure.
- The sections Antioxidant Activity, Cellular Neuroprotective Assay, and Zebrafish Embryo Toxicity Assay are incorrectly labeled as 2.8, 2.3.2, and 3.4. These should be corrected according to the numbering in the paper. Check the section numbering throughout the document.
This manuscript is scientifically sound, contains 77 references, is relevant and novel. The conclusions are consistent with the evidence and arguments provided. The ethical statements and statements about the availability of data are adequate.
In my opinion, the manuscript can be accepted for publication in the journal after minor additions and corrections.
The manuscript on the topic "Potential antioxidant and neuroprotective effect of polysaccharide isolated from digüeñe Cyttaria espinosae" refers to experimental articles, is clearly and well-structured. The authors conducted a study of the antioxidant and neuroprotective effects of polysaccharides from Cyttaria espinosae on neurotoxicity induced by beta-amyloid (Aβ) in neuron-like cells. The toxicity of this polysaccharide was assessed in vivo in zebrafish embryos. It was found that CePs have antioxidant and neuroprotective effects and are non-toxic to zebrafish embryos.
The results of this study are the basis for further study of properties in the context of the physiopathology of Alzheimer's disease.
There are several minor comments on the article:
- In section 2.2.2. provide in detail the method for identifying Cyttaria espinosae samples.
- Sections 3.2.1 and 3.3.3 need to be expanded. Provide the calibration curve by which the total sugar content in CeP was determined.
- In Figure 4, it is advisable to indicate the control (ascorbic acid) and provide explanations for A, B, C, D, E in the figure.
- The sections Antioxidant Activity, Cellular Neuroprotective Assay, and Zebrafish Embryo Toxicity Assay are incorrectly labeled as 2.8, 2.3.2, and 3.4. These should be corrected according to the numbering in the paper. Check the section numbering throughout the document.
This manuscript is scientifically sound, contains 77 references, is relevant and novel. The conclusions are consistent with the evidence and arguments provided. The ethical statements and statements about the availability of data are adequate.
In my opinion, the manuscript can be accepted for publication in the journal after minor additions and corrections.
Author Response
Reviewer 2
The manuscript on the topic "Potential antioxidant and neuroprotective effect of polysaccharide isolated from digüeñe Cyttaria espinosae" refers to experimental articles, is clearly and well-structured. The authors conducted a study of the antioxidant and neuroprotective effects of polysaccharides from Cyttaria espinosae on neurotoxicity induced by beta-amyloid (Aβ) in neuron-like cells. The toxicity of this polysaccharide was assessed in vivo in zebrafish embryos. It was found that CePs have antioxidant and neuroprotective effects and are non-toxic to zebrafish embryos.
The results of this study are the basis for further study of properties in the context of the physiopathology of Alzheimer's disease. There are several minor comments on the article:
- In section 2.2.2. provide in detail the method for identifying Cyttaria espinosae
Thank you for your valuable feedback and suggestions regarding our manuscript. We have taken your comments into account and made several improvements in section 2, "Materials and Methods," specifically in section 2.2.2, where we provide a detailed account of the identification methods employed for Cyttaria espinosae samples.
In this revised section, we have incorporated relevant references to enhance scientific rigor and support our methods and findings more effectively. The addition of these references not only provides a broader context for our work but also aligns our methodology with current literature in the field. We appreciate your attention to this detail, and we will make sure this clarification is clearly articulated in the manuscript.
2.2.2. Identification
In the identification of Cyttaria espinosae samples, we carefully documented the macroscopic characteristics of the stromata in situ, utilizing photographic documentation via a Canon T6 reflex camera for both field and laboratory observations. This step is vital for establishing a reliable morphological baseline for identification purposes. Taxonomic determination was then achieved through an examination of microscopic features. Stromata morphology was examined using a Motic BA210S Trinocular compound microscope (Barcelona, Spain) with a Moticam S6 camera attached. Microscopic structures were studied by mounting tissue sections of dried stromata in demineralized water and lactophenol cotton blue according to established methods [16].
Post-identification, we ensured that the stromata were labeled meticulously and subsequently frozen at -20°C. Following freezing, the stromata underwent lyophilization for 48 hours using a CHRIST Alpha 2-4 LO plus automatic lyophilizer. This process enhances the long-term viability of the samples and prevents structural degradation caused by the formation of ice crystals [17]. The final step involved storing the lyophilized specimens at -20°C to prepare them for future analytical research on their nutritional and bioactive properties [18].
The references included are as follows:
[17]. Villalobos-Pezos, J.; González, P.; Córdova, E.; Duran, R.; Antezana, M. Effects of Drying Treatments on the Physicochemical Characteristics and Antioxidant Properties of the Edible Wild Mushroom Cyttaria espinosae Lloyd. Journal of Fungi 2024, 10, 80. doi:10.3390/jof11010013.
[18]. Sorokulova, I.; Lantukh, D.; Kruchinina, M.; Sabitov, A. Biopolymers for sample collection, protection, and preservation. Applied Microbiology and Biotechnology 2015, 99, 5995–6003. doi:10.1007/s00253-015-6681-3.
- Sections 3.2.1 and 3.3.3 need to be expanded. Provide the calibration curve by which the total sugar content in CeP was determined.
Thank you for your insightful feedback regarding the need to expand Sections 3.2.1 and 3.3.3. In these sections, we provided a detailed methodology for determining the total sugar content in the crude extracts of fruiting bodies (CeP).
3.2.1. Yield of Isolation of Fruiting Body Polysaccharides
From a total of 74.41 g of lyophilized fruiting body of C. espinosae, we successfully isolated 6.48 g of crude polysaccharides (CePs), representing an extraction yield of 8.71% (see Table 1). This yield was achieved through a series of well-established methods. Initially, the biomass was treated with 70% ethanol for 48 hours to remove impurities. Subsequently, polysaccharides were extracted by heating the biomass in distilled water at 100°C, followed by centrifugation to recover the supernatant twice for thorough extraction. The supernatant was concentrated using a rotary evaporator and precipitated by adding absolute ethanol in a 1:1 ratio, ultimately leading to the collection of crude polysaccharides that were lyophilized for storage. This yield is consistent with typical extraction rates observed in various fungal species.
3.3.3 Total Phenolic Content
The total phenolic content (TPC) in the polysaccharides extracted from Cyttaria espinosae was assessed using the Folin-Ciocalteu colorimetric method. The TPC measured was 0.801 ± 0.34 mg GAE g⁻¹ DW (gallic acid equivalents per gram of dry weight).
Phenolic compounds are recognized for their antioxidant properties and numerous health benefits, underscoring the importance of this measurement. While the Folin-Ciocalteu method is widely utilized for its effectiveness in quantifying total phenolic compounds, it is essential to note that other substances can interfere with the results, potentially leading to overestimations of TPC [41].
Moreover, the TPC observed in this study contributes to the body of evidence supporting the nutritional and medicinal potential of CePs. Further investigation into the specific phenolic compounds present, as well as their biological activity, could provide valuable insights regarding their functional applications.
[41]Bunzel, M.; Schendel, R.R. Determination of (Total) Phenolics and Antioxidant Capacity in Food and Ingredients. In Food Analysis; Nollet, L.M.L., Toldrá, F., Eds.; Springer: Cham, Switzerland, 2017; pp. 55–468. https://doi.org/10.1007/978-3-319-45776-5_25
Provide the calibration curve by which the total sugar content in CeP was determined.
To determine the total sugar content in the polysaccharides of Cyttaria espinosae (CePs), we employed the modified phenol-sulfuric acid method, generating a calibration curve using glucose as the standard (Figure). While this curve was not initially included in the manuscript, we appreciate your suggestion and will add it to the supplementary materials to enhance the methodological clarity of our study.
Supplementary Material S1:

Figure. Calibration curve of glucose
- In Figure 4, it is advisable to indicate the control (ascorbic acid) and provide explanations for A, B, C, D, E in the figure.
In response to your suggestion regarding Figure 4, we appreciated your valuable input and made the necessary revisions to improve the clarity and comprehensibility of the figure. Specifically, we included an indication of the control (ascorbic acid) within the figure, which served as a baseline for comparison. Additionally, we provided clear explanations for the labels A, B, C, D, and E to ensure that readers could easily understand the findings presented.
The changes implemented in the revised figure included:

Figure 4. Scavenging effects (%) of the crude polysaccharides from Cyttaria espinosae (CePs) on DPPH and ABTS radicals at varying concentrations. The results are expressed as the mean of three replicate measurements, with error bars representing the standard error. Analysis of Variance (ANOVA) was performed, and different lowercase letters indicate significant differences between samples at the 0.05 level.
The sections Antioxidant Activity, Cellular Neuroprotective Assay, and Zebrafish Embryo Toxicity Assay are incorrectly labeled as 2.8, 2.3.2, and 3.4. These should be corrected according to the numbering in the paper. Check the section numbering throughout the document.
Thank you for bringing the issue of incorrect section numbering to our attention regarding the sections titled "Antioxidant Activity," "Cellular Neuroprotective Assay," and "Zebrafish Embryo Toxicity Assay." We conducted a thorough review of the manuscript to ensure all sections were correctly labeled according to the appropriate numbering sequence.
In the revisions, we:
Corrected the section number for "Antioxidant Activity" from 2.8 to the appropriate number as per the manuscript's structure.
Updated the section number for "Cellular Neuroprotective Assay" from 2.3.2 to align with the correct order.
Amended the section number for "Zebrafish Embryo Toxicity Assay" from 3.4 to the designated number in the context of the document.
Line 219 2.10 Antioxidant Activity
Line 220 2.10.1 Scavenging Ability on 1,1-Diphenyl-2-picrylhydrazyl Radicals
Line 238 2.10.2 ABTS Radical Scavenging Capacity Assay
Line 258 2.10.3 Total Antioxidant Capacity (TEAC)
Line 267 2.11 Biological Assays
Line 268 2.11.1 PC12 Cells
Line 276 2.11.2 Cytotoxicity Assay (MTT)
Line 295 2.11.3 Aggregation of Soluble Oligomers of the Aβ1-40 Peptide
Line302 2.11.4 Neuroprotection Assay Against Aβ-Generated Toxicity (MTT)
Line 309 2.12 Zebrafish Embryo Toxicity Assay
Line 339 2.13 Statistical Analysis
Additionally, we carefully checked the numbering of all sections throughout the manuscript to ensure consistency and accuracy.
We appreciate your guidance in this matter, which has improved the overall clarity and organization of the manuscript.
Reviewer 3 Report
This article investigates the potential of a novel microbial-derived polysaccharide in treating Alzheimer's disease. The writing is standardized and the data supports the conclusions.
- Line 93-95: Please provide additional information about monosaccharide purity.
- Line 77-79: Similar to the previous point, chemical reagent information is insufficient.
- Line 104: Is this species wild? Can it only be collected in spring?
- Line 124: Equipment specifications are missing.
- Line 174: Why was ATR mode used for infrared measurement? This mode is typically for film analysis.
- Table 1 lacks standard deviations for several data sets.
- Table 2: Are these mean values? Standard deviations are absent.
- Figure 2 has low resolution.
- Section 3.2.4: Peak assignments require supporting references.
Author Response
Reviewer 3
Major comments
This article investigates the potential of a novel microbial-derived polysaccharide in treating Alzheimer's disease. The writing is standardized and the data supports the conclusions.
Dear Reviewer,
Thank you for your positive feedback on our manuscript investigating the potential of Cyttaria espinosae polysaccharide in treating Alzheimer's disease.
Detailed comments
- Line 93-95: Please provide additional information about monosaccharide purity.
We have added additional information regarding the purity of the monosaccharides used in our study. The revised sentence in lines 93-97 now reads:
“Standard monosaccharides, including L-arabinose (Ara), erythrose (Ery), L-fucose, D-galactose (Gal), D-glucose (Glc), D-mannose (Man), L-rhamnose (Rha), D-ribose (Rib), and D-xylose (Xyl), were obtained from Sigma Chemical Co. (St. Louis, MO, USA). All monosaccharides were of analytical grade with a purity ≥99.5% (GC), as specified by the supplier.”
- Line 77-79: Similar to the previous point, chemical reagent information is insufficient.
Thank you for pointing this out. We have revised the manuscript to include more detailed information about the chemical reagents used. The updated section now reads:
“Additionally, chemical reagents were obtained from Sigma–Aldrich (St. Louis, MO, USA). Standards of gallic acid (≥98% purity), Trolox (≥98% purity, titration) 1, 2,2-diphenyl-1-picrylhydrazyl (DPPH•⁺, ≥95% purity), and ascorbic acid (≥99% purity) were of analytical grade and used without further purification. Ethanol (≥99.8%, analytical grade), methanol (≥99.9%, analytical grade), and the Folin–Ciocalteu reagent were supplied by Merck (Darmstadt, Germany).”
- Line 104: Is this species wild? Can it only be collected in spring?
In response to your inquiry regarding the wild nature of Cyttaria espinosae and its collection period, we would like to clarify that Cyttaria espinosae is indeed a wild species. This edible fungus typically fruits during the spring and summer seasons, which is when we conducted our collection efforts in the Nahuelbuta area in 2017-2018.
Although our study primarily focused on the spring-summer season for optimal collection, it is important to note that the fruiting of Cyttaria espinosae may vary depending on environmental conditions and specific habitats. Therefore, while the species is most reliably collected during this period, it could potentially be found at other times, albeit with less frequency or in smaller quantities.
We appreciate your attention to this detail, and we will make sure this clarification is clearly articulated in the manuscript.
- Line 124: Equipment specifications are missing.
Thank you for your comment. We have now included the equipment specifications in the revised manuscript. The updated sentence in line 124 reads:
“The lyophilized sample of C. espinosae was crushed using a Moulinex DP800 grinder (Moulinex, France), which operates at 1000 W with stainless steel blades, suitable for fine powder preparation. The initial mass of the sample was recorded using a precision balance (±0.01 g).”
- Line 174: Why was ATR mode used for infrared measurement? This mode is typically for film analysis.
In response to your question regarding the use of Attenuated Total Reflection (ATR) mode for the infrared measurement of polysaccharide extracts, we appreciate your inquiry. ATR-FTIR spectroscopy was employed in our study due to its advantages in analyzing solid samples, particularly those with limited thickness, such as polysaccharide extracts.
ATR mode enables minimal sample preparation and provides high sensitivity when measuring the infrared spectra of complex samples. This technique is particularly beneficial for characterizing the chemical composition of polysaccharides, as it enables the collection of high-quality spectra without requiring extensive sample manipulation or dilution, which could otherwise compromise the results (Giuliano et al., 2020). Moreover, the high refractive index of the ATR crystal enhances the resolution of the spectral data, allowing for a detailed analysis of the functional groups present in the polysaccharides (Wróbel et al., 2012).
Specifically, we will add this information to clarify the rationale behind the choice of ATR mode for analyzing polysaccharide extracts. The updated sentence in line reads:
“The Attenuated Total Reflection (ATR) mode for FT-IR spectroscopy, due to its advantages in analyzing solid samples like polysaccharide extracts, allows for minimal sample preparation and high sensitivity, which is crucial for accurately characterizing their chemical composition [22, 23] (Giuliano et al., 2020; Wróbel et al., 2012).
References add:
22. Giuliano, S., Mistek, E., & Lednev, I. K. (2020). Forensic phenotype profiling based on the attenuated total reflection fourier transform-infrared spectroscopy of blood: chronological age of the donor. ACS Omega, 5(42), 27026-27031. https://doi.org/10.1021/acsomega.0c01914
23. Wróbel, T. P., Marzec, K. M., Majzner, K., Kochan, K., Bartuś, M., Chłopicki, S., … & Barańska, M. (2012). Attenuated total reflection fourier transform infrared (atr-ftir) spectroscopy of a single endothelial cell. The Analyst, 137(18), 4135. https://doi.org/10.1039/c2an35331h
- Table 1 lacks standard deviations for several data sets.
Thank you for your observation. We have revised Table 1 to include standard deviations for all applicable data sets. These values are now presented as mean ± standard deviation(SD), based on triplicate measurements
- Table 2: Are these mean values? Standard deviations are absent.
Table 2. Total Carbon (TC), Total Nitrogen (TN), Total Hydrogen (TH), and Total Sulphur (TS) content of CePs
Elemental composition of Cyttaria espinosae polysaccharide (%)
TC
TH
TN
TS
37.85±0.5
6.593±0.2
0.679±0.09
0.000±0.02
n = 3 (represents the average of three measurements from each point)
According to the obtained molar (C/N ratio) value was 65.03 in polysaccharides
Thank you for pointing this out. Yes, the values in Table 2 represent means. We have now clarified this in the table caption and added the corresponding standard deviations (mean ± SD) to ensure consistency and transparency.
- Figure 2 has low resolution.
Thank you for your comment. Figure 2 has been replaced with a high-resolution version (minimum 300 dpi) to ensure clarity and readability in the final publication

- Section 3.2.4: Peak assignments require supporting references.
Thank you for your valuable comment regarding the need for supporting references for the FT-IR peak assignments in Section 3.2.4. We would like to clarify that the rationale and literature supporting the peak assignments have already been discussed in detail in the Discussion section of the manuscript. To improve clarity and guide the reader, we will add a sentence in lines of Section 3.2.4, explicitly referring to the corresponding discussion and references.
The updated sentence reads:
Line 436-438 These assignments are consistent with previously reported FT-IR profiles of polysaccharides and carbohydrates [38–40].
Round 2
Reviewer 1 Report
Not more comments.
Not more comments.
Author Response
Dear Reviewer
We sincerely thank you and the reviewers for the constructive feedback provided on our manuscript titled “Potential antioxidant and neuroprotective effect of polysaccharide isolated from digüeñe Cyttaria espinosae” (Manuscript ID: jof-3723030).
We are pleased to inform you that all suggestions and comments have been carefully considered and incorporated into the revised version of the manuscript. In particular, we have addressed the following key points:
- Reduction of repeated sentences and ideas throughout the manuscript.
- Reorganization of the Results section to remove methodological and interpretative content.
- Streamlining of the Discussion section to focus on the most relevant aspects aligned with the study’s aim.
- Clarification and consistency in data presentation and terminology.
- Inclusion of additional methodological details and corrections as requested.
For your convenience, we have attached a detailed document titled “Reply to Reviewer 2025”, which outlines our responses to each comment and the corresponding revisions made.
We appreciate your guidance and support throughout the review process and look forward to your feedback on the revised submission.
Kind regards,
Dra. Claudia Pérez Manríquez
Laboratorio de Química de Productos Naturales
Departamento de Botánica
Facultad de Ciencias Naturales y Oceanográficas
Universidad de Concepción
ORCID: 0000-0002-8562-3842
Email: claudiaperez@udec.cl
